# Pharmacokinetic Analysis of Levodropropizine and Its Potential Therapeutic Advantages Considering Eosinophil Levels and Clinical Indications

**DOI:** 10.3390/ph17020234

**Published:** 2024-02-10

**Authors:** Ji-Hun Jang, Young-Jin Cho, Seung-Hyun Jeong

**Affiliations:** 1College of Pharmacy, Sunchon National University, 255 Jungang-ro, Suncheon-si 57922, Jeollanam-do, Republic of Korea; jangji0121@naver.com (J.-H.J.); 20226024@s.scnu.ac.kr (Y.-J.C.); 2College of Pharmacy and Research Institute of Life and Pharmaceutical Sciences, Sunchon National University, Suncheon-si 57922, Jeollanam-do, Republic of Korea

**Keywords:** levodropropizine, pharmacokinetics, inter-individual variability, covariates, body surface area, eosinophil levels

## Abstract

Levodropropizine is a non-narcotic, non-centrally acting antitussive that inhibits the cough reflex triggered by neuropeptides. Despite the active clinical application of levodropropizine, the exploration of its inter-individual pharmacokinetic diversity and of factors that can interpret it is lacking. The purpose of this study was to explore effective covariates associated with variation in the pharmacokinetics of levodropropizine within the population and to perform an interpretation of covariate correlations from a therapeutic perspective. The results of a levodropropizine clinical trial conducted on 40 healthy Korean men were used in this pharmacokinetic analysis, and the calculated pharmacokinetic and physiochemical parameters were screened for effective correlations between factors through heatmap and linear regression analysis. Along with basic compartmental modeling, a correlation analysis was performed between the model-estimated parameter values and the discovered effective candidate covariates for levodropropizine, and the degree of toxicity and safety during the clinical trial of levodropropizine was quantitatively monitored, targeting the hepatotoxicity screening panel. As a result, eosinophil level and body surface area (BSA) were explored as significant (*p*-value < 0.05) physiochemical parameters associated with the pharmacokinetic diversity of levodropropizine. Specifically, it was confirmed that as eosinophil level and BSA increased, levodropropizine plasma exposure increased and decreased, respectively. Interestingly, changes in an individual’s plasma exposure to levodropropizine depending on eosinophil levels could be interpreted as a therapeutic advantage based on pharmacokinetic benefits linked to the clinical indications for levodropropizine. This study presents effective candidate covariates that can explain the inter-individual pharmacokinetic variability of levodropropizine and provides a useful perspective on the first-line choice of levodropropizine in the treatment of inflammatory respiratory diseases.

## 1. Introduction

Chronic cough significantly reduces the quality of human life and interferes with daily life. Therefore, the prompt treatment of chronic cough and the selection of appropriate medications are important aspects of clinical practice. Unfortunately, many cough medications offer limited relief for most patients [1]. Levodropropizine stands out as an effective antitussive for both adults and children, demonstrating notable reductions in cough intensity, frequency, and nighttime awakenings compared to centrally acting antitussives [2]. While centrally acting antispasmodics are typically narcotic, posing potential central nervous system (CNS) side effects [3], levodropropizine is a non-opioid peripheral nerve agent. It has been administered for years, even to children as young as two years old [3]. Its exact pharmacological mechanism remains unclear, but it is believed to exert its antitussive effects by inhibiting vagal C-fiber activation [4]. It was superior to placebo in cough caused by bronchitis, comparable to dextromethorphan in patients with dry cough, and similar to codeine in cough associated with lung cancer [5,6]. As a result, levodropropizine is relatively free from CNS side effects such as drowsiness, which is the biggest side effect of cough medications, and has the therapeutic advantage of being applicable to wide-ranging groups, from young children to adults. In addition, the fact that it is not at all inferior to other cough medications in terms of efficacy suggests that levodropropizine would be the preferred choice for the treatment of chronic cough in clinical practice.

Levodropropizine is mainly marketed in oral dosage forms, and injection dosage forms have not been identified. Orally administered levodropropizine is rapidly absorbed through the gastrointestinal tract, with the time required to reach maximum drug concentration (T_max_) ranging from 0.25 to 2 h. In the low-pH environment of the stomach (approximately pH 1.0 to 3.0), levodropropizine exists in its 100% ionized form, has high bioavailability (75%), and is rapidly distributed [7]. The reported mean elimination half-life (T_1/2_) of levodropropizine is 2.3 h [8], suggesting rapid elimination after absorption in the body. Levodropropizine is reported to have a linear pharmacokinetic profile in the dose range of 30 to 90 mg [9]. As a result, levodropropizine is judged to have relatively good absorption and distribution in the body, considering the previously reported pharmacokinetic aspects. In addition, despite frequent exposure, such as three times a day (for 5 days) in the general dosage form (60 mg tablet), it is expected that there is no significant in vivo accumulation and that there are no atypical pharmacokinetic phenomena. Levodropropizine has been found to be well tolerated [6] and has been shown to be safe at doses up to 10 times (24 mg/kg/day) the clinical dose in rats and mice, with only mild adverse events at 30 times (approximately 80 mg/kg/day) the clinical dose [10].

A precision medicine approach is important to optimize cough treatment with levodropropizine, and a key step toward this is the exploration of effective covariates that can account for inter-individual pharmacokinetic variations. To date, levodropropizine clinical studies have primarily focused on the bioequivalence of generic or test formulations [11,12,13,14], with only one considering its pharmacokinetic parameters in relation to food intake [8]. For instance, a report by Jang et al. [11] primarily interpreted existing datasets (without exploring hidden covariate factors), providing a narrow scope of pharmacokinetic analysis. As a result, existing reports [8,11,12,13,14] on levodropropizine have limited utility in exploring individualized clinical treatment factors and/or dose adjustments. Recently, a population pharmacokinetic study on levodropropizine was reported, focusing on differences in formulation and dietary factors [15]. However, in the previous study [15], the exploration of covariates linked to aspects of treatment effect related to clinical indications (such as eosinophilic asthma) in interpreting the pharmacokinetic diversity of levodropropizine was not performed. This suggested the need for further expanded covariate explorations that closely link levodropropizine pharmacokinetic variability interpretations with clinical-indication-related factors. Although levodropropizine is generally well tolerated, with temporary nausea as its only reported mild side effect [6], it is very important to interpret the phenomenon of pharmacokinetic variability between individuals and identify relevant key factors. This is because it not only helps to quantitatively predict the treatment effect from a scientific approach, but also maximizes effectiveness and safety information to an unprecedented extent by optimizing application. Still, dosing decisions in clinical settings are based on empirical knowledge, and there is a conspicuous absence of studies aimed at uncovering covariates that might influence levodropropizine’s pharmacokinetic parameters or translating these findings into clinical dosage adjustments. Given this backdrop, our study seeks to build upon foundational pharmacokinetic diversity correlation factor discovery research.

We aim to ascertain whether certain physiochemical traits, pivotal for individualized pharmacotherapy, can serve as reliable covariates for levodropropizine pharmacokinetics. And ultimately, we sought to explore rational judgments that could link the physiochemical factors explored as candidate covariates in this study to the therapeutic benefits of levodropropizine. The results of this study will serve as scientific data that narrow the gap in existing knowledge about the pharmacokinetics and inter-individual variability of levodropropizine, and will also serve as a useful reference in performing advanced pharmacometrics modeling (through the addition of candidate covariates such as eosinophil levels that may be more closely related to therapeutic aspects).

## 2. Results

### 2.1. Clinical Physiochemical Parameter Analysis

The participants’ age, height, and weight ranged from 19 to 45 years, 161.8 to 188.7 cm, and 55.4 to 91.9 kg, respectively. Their body surface area (BSA) and body mass index (BMI) are 1.57–2.15 m^2^ and 19.1–29.1 kg/m^2^, respectively. The concentrations of white blood cells, red blood cells, and platelets ranged from 4.58 to 10.14 (×10^3^ count/μL), 4.49 to 6.51 (×10^6^ count/μL), and 178 to 342 (×10^3^ count/μL), respectively. Hemoglobin concentration ranged from 13.1 to 17.8 g/dL and hematocrit concentration ranged from 41.8 to 54.4%. The levels of neutrophils, lymphocytes, and eosinophils ranged from 40.2 to 78.5%, 15.8 to 44.8%, and 0.4 to 10.6%, respectively, and absolute neutrophil count concentrations ranged from 1.89 to 6.89 (×10^3^ count/mm^3^). The levels of blood urea nitrogen, creatinine, total protein, and albumin were in the ranges of 7.6–22.3 mg/dL, 0.7–1.1 mg/dL, 6.8–7.9 g/dL, and 4.3–5.1 g/dL, respectively. The values of estimated glomerular filtration rate (eGFR) and creatinine clearance (CrCL) calculated based on the creatinine value were in the range of 77.6–143.8 mL/min/1.73 m^2^ and 76.9–166.7 mL/min, respectively. The levels of alkaline phosphatase (ALP), aspartate transaminase (AST), alanine transaminase (ALT), and gamma-glutamyl transpeptidase (γ-GTP) were in the ranges of 46–104 U/L, 17–40 U/L, 10–78 U/L, and 8–71 U/L, respectively. Total bilirubin, glucose, and total cholesterol concentration values were in the ranges of 0.5–2.5 mg/dL, 73–96 mg/dL, and 116–273 mg/dL, respectively.

### 2.2. Pharmacokinetic Studies and Analysis

T_1/2_ of levodropropizine has been reported to be approximately 2.3 h [8]. Therefore, the clinical protocol design for this pharmacokinetic study set the sampling time after administration to 12 h, which is more than four times the T_1/2_. And based on reports that the T_max_ of levodropropizine is approximately within 0.25–2 h [8,11,12,13,14], more than 50% of all sampling points were tightly (at intervals of 10 to 30 min) set within 2 h after administration. This was to properly capture peak levodropropizine concentration in plasma (C_max_) and T_max_ due to the expected rapid oral absorption of levodropropizine. As a result of the non-compartmental analysis of levodropropizine, area under the curve (AUC) from 0 h to last observation time point (AUC_0-t_) and AUC from 0 h to infinity (AUC_0-inf_) values were 955 ± 380 and 1000 ± 403 h·ng/mL (mean ± standard deviation [SD]), respectively. And the average ratio between AUC_0-t_ and AUC_0-inf_ values was high, approximately 95.5%, suggesting that the sampling protocol established in this clinical study was appropriate to minimize extrapolated AUC. Furthermore, a lower limit of quantification of 5 ng/mL (as the analytical sensitivity established in this study) was suggested to be sufficiently sensitive for pharmacokinetic studies on plasma samples obtained after oral administration of 60 mg levodropropizine tablet to humans. C_max_ was 452 ± 203 ng/mL (mean ± SD) and T_max_ was very short at 0.6 ± 0.3 h (mean ± SD). This suggested rapid oral absorption of levodropropizine, consistent with previous reports [8,11,12,13,14]. T_1/2_ was 2.3 ± 0.5 h (mean ± SD), mean residence time (MRT) was 3.2 ± 0.6 h (mean ± SD), and clearance (CL/F) was 69 ± 26 L/h (mean ± SD), suggesting rapid and extensive elimination of levodropropizine from the body. The volume of distribution (V/F) was 211 ± 46 L (mean ± SD), suggesting an extensive distribution of levodropropizine throughout the body, especially to peripheral tissues. The plasma concentrations of levodropropizine versus time profiles for 40 healthy Korean subjects are shown in Figure 1. As confirmed in the results of pharmacokinetic parameter analysis through non-compartmental analysis, levodropropizine had a rapid oral absorption (within 10 min after administration) and relatively variable absorption patterns.

### 2.3. Safety Screening

The values of AST, ALT, and γ-GTP selected as panels for the liver toxicity test were compared at two points: before (0 h) and 12 h after levodropropizine administration. Figure 2 shows the changes in levels of AST, ALT, and γ-GTP in a spaghetti plot (as a visual representation using a continuous flow line tracing the path of an item). Large increases in AST, ALT, and γ-GTP levels may suggest liver damage secondary to drug exposure. However, according to the safety screening analysis results confirmed in this study, no notable increases in AST, ALT, and γ-GTP levels were confirmed. In other words, most of the fluctuations in liver toxicity panels observed after single oral administration of levodropropizine were within 20% of baseline values (as measured at 0 h). And even if there was an increase of more than 10% from baseline, these values fell within a reasonable normal range. As a result, it was confirmed that no significant hepatotoxic side effects would occur following single oral exposure to levodropropizine 60 mg tablet in healthy adult populations. Even during single oral exposure to levodropropizine 60 mg tablet and subsequent follow-up monitoring, no additional side effects (from mild gastrointestinal symptoms to major hypersensitivity) were observed. This suggested a favorable safety profile of levodropropizine.

### 2.4. Covariate Screening Results

The results of heatmap analysis performed by screening for correlations between pharmacokinetic parameter values obtained through non-compartmental analysis and physiochemical parameters are presented in Figure 3. Eosinophil levels and BSA were explored as physiochemical factors with valid correlations (correlation coefficient is greater than +0.3 or less than −0.3) with pharmacokinetic parameters. On the other hand, other physiochemical factors were not valid for screening correlation with pharmacokinetic parameters. In particular, no significant correlations with pharmacokinetic parameters were identified for eGFR, creatinine, and CrCL, which are used as indicators of renal function, and ALP, AST, ALT, and γ-GTP, which are generally used as indicators of liver function.

Figure 4 shows the linear regression analysis results for elements whose coefficient values are at the top of the valid correlation derived from the heatmap screening results and for which grounds for correlation interpretation have been secured. All regression analyzes showed a correlation (with *R*-squared value greater than 0.09 [16]) of more than 30% between variables, and the *p*-value was statistically significant at less than 0.05. Notably, regression analysis revealed a significant correlation between eosinophil levels and various pharmacokinetic parameters of levodropropizine. As eosinophil levels increased, levodropropizine’s C_max_, T_1/2_, and AUC_0-t_ were confirmed to increase in positive correlation, while V/F and CL/F were confirmed to be decreased in negative correlation. And the overall trend of changes in pharmacokinetic parameters according to eosinophil levels was consistent, with the plasma exposure of levodropropizine increasing as eosinophil levels increased.

In the individual comparison of levodropropizine pharmacokinetic profiles of subjects with the highest (as 10.6%) and lowest (as 0.4%) eosinophil levels in this clinical study’s population, the pattern of increased levodropropizine exposure as the eosinophil level increased was clearly confirmed. Figure 5 shows a comparison of levodropropizine pharmacokinetic profiles in subjects with the highest and lowest eosinophil levels. All comparison subjects were healthy adults, and no large differences were observed in liver and renal function values. That is, in the case of the individual with high eosinophil levels, liver function-related values of AST, ALT, and γ-GTP were 23 U/L, 13 U/L, and 15 U/L, respectively, and the renal function-related value of creatinine was 1.1 mg/dL. On the other hand, in the individual with low eosinophil levels, AST, ALT, and γ-GTP were 18 U/L, 18 U/L, and 15 U/L, respectively, and creatinine had a value of 0.9 mg/dL. Compared to the subject with the lowest eosinophil level, C_max_, T_1/2_, MRT, and AUC_0-t_ in the subject with the highest eosinophil level were increased by 2.12 (from 569.14 to 1205.95 ng/mL), 1.23 (from 2.56 to 3.14 h), 1.25 (from 3.14 to 3.92 h), and 2.48 (from 872.52 to 2167.80 h·ng/mL) times, respectively. And V/F and CL/F were reduced by 2.08 (from 205.14 to 98.66 L) and 2.55 (from 55.53 to 21.75 L/h) times, respectively. Additionally, notable trends were identified between BSA and levodropropizine pharmacokinetic parameters, and overall, as BSA increased, levodropropizine exposure in plasma decreased (with a decrease in C_max_ and AUC and an increase in CL/F). In particular, the tendency for V/F to increase as BSA increased showed a high correlation, with an *R*-squared value of 0.14.

### 2.5. Compartmental Model Screening

As a result of trying several models to screen the basic model structure, a two-compartment model with a lag time was the most suitable for levodropropizine pharmacokinetics and the selected model correlated even better with eosinophil levels. The two-compartment model with a lag time exhibited the lowest negative Akaike information criterion (AIC) value of −24.26, indicating a favorable balance between model fit and complexity when compared to other models. Similarly, the Schwarz criterion (SC) was also the lowest at −21.35. The model’s precision was further evidenced by the weighted sum of squared residuals (WSSR), which was the smallest observed at 0.05 (as a mean value), suggesting the least disparity between the model predictions and the actual observed values. Figure 6 shows the model fitting results for the observed mean plasma concentration values and WSSR distribution patterns. Furthermore, the run test (Figure 6; as the total number of times the WSSR connection line of all points passes 0), which assesses the consistency of the model and the randomness of data fluctuations by counting the number of times the data points deviate above or below the predicted values, showed the highest count (of 6) in the selected model among the tried models. Additionally, the two-compartment model with a lag time could be applied to all subjects without non-fitting error. As a result, it was confirmed that the two-compartment model with a lag time would be appropriate as a basic structure to explain the pharmacokinetics of levodropropizine in healthy Korean men.

The C_max_, T_1/2_, and AUC_0-inf_ estimates derived from the two-compartment model with a lag time underwent additional linear regression analysis against eosinophil levels, for which a correlation had been identified based on the results of non-compartmental analysis (Figure 4). Figure 7 shows the results of regression correlation analysis between pharmacokinetic parameters (C_max_ and AUC_0-t_) estimated from the basic structure of the compartment model explored in this study and eosinophil levels. It was confirmed that the tendency for plasma exposure to levodropropizine to increase due to increased eosinophil levels was the same in the application of the screened compartmental model as in the non-compartmental model analysis. All correlations were statistically significant (*p*-value lower than 0.05) and overall were higher than the correlations (higher *R*-squared) between pharmacokinetic parameters (C_max_ and AUC_0-t_) calculated by non-compartmental analysis and eosinophil levels. This suggested the possibility that the eosinophil level explored in this study could be established as a powerful factor that can explain the inter-individual pharmacokinetic variability of levodropropizine in future expanded pharmacokinetic modeling studies (including quantitative covariate reflection mathematically).

## 3. Discussion

The pharmacokinetic results identified in this study were not largely different compared to previous reports [11,12,13,14]. For example, according to a recent report, the mean values of AUC_0-inf_, T_1/2_, and T_max_ following oral administration of immediate-release levodropropizine tablet (60 mg, in fasted state) were 969.06 h·ng/mL, 2.30 h, and 0.75 h, respectively, which were similar to the values of 1000 h·ng/mL, 2.3 h, and 0.6 h determined in this study. This implied that the pharmacokinetic results of levodropropizine derived from this study were not largely different from those in previous reports [8,11,12,13,14] and provided consistent and useful information on the pharmacokinetic properties of levodropropizine in humans.

The most important and interesting point of this study was the exploration of the correlation between the pharmacokinetic parameters of levodropropizine and eosinophil levels. This is because maintaining a high plasma concentration of levodropropizine is a factor that can be a great advantage in therapeutic terms, and (in this study) as the level of eosinophils increases, the level of levodropropizine plasma exposure in individuals significantly increases. Considering that antitussive drugs such as levodropropizine are generally used to alleviate inflammatory respiratory diseases [3], levodropropizine may be judged to be a relatively preferable medicine compared to other drugs in the same class in terms of pharmacokinetics. That is, in this clinical study, in subjects who had relatively increased eosinophil levels (before levodropropizine administration), levodropropizine plasma exposure increased following administration, which may result in a therapeutic advantage (based on the general relationship of drug efficacy proportional to drug concentration in plasma). The positive correlation between eosinophil levels and levodropropizine exposure in plasma can be interpreted in relation to the decrease in metabolism in vivo due to inflammation. It would be possible that inflammation reduces hepatic first-pass metabolism of levodropropizine and concurrently increases eosinophil levels, serving as an indicator of inflammation. Previous studies have demonstrated that inflammation can raise eosinophil levels [17] and impair liver metabolic processes [18,19]. Additionally, inflammation is known to suppress the expression and activity of cytochrome P450 enzymes [20], which play a crucial role in hepatic drug metabolism, and can also influence the regulation of human drug metabolizing enzymes (such as esterase and phase II-related enzymes) and transporters, resulting in varied drug responses and potential toxicity (with significant pharmacokinetic changes) [21]. In addition, high eosinophil levels could themselves contribute to inflammation, as seen in eosinophilic asthma where eosinophils act as pro-inflammatory agents and can suppress hepatic first-pass metabolism. In this context, levodropropizine might exhibit enhanced effectiveness, particularly in eosinophilic asthma, a common variant of asthma [22,23]. Elevated eosinophil levels have been associated with an increased risk of severe asthma attacks and challenges in symptom management [24], suggesting that levodropropizine may be a beneficial antitussive agent in the treatment of severe eosinophilic asthma. Since this study was a retrospective pharmacokinetic analysis using bioequivalence results for levodropropizine, it will be necessary to conduct additional prospective pharmacokinetic and pharmacodynamic studies in the future considering the correlation between eosinophil levels and levodropropizine pharmacokinetics. This will enable a quantitative scientific approach to antitussives based on the pharmacometrics of levodropropizine and make it possible to more clearly determine the effectiveness of covariates in the clinical application of levodropropizine. In particular, through pharmacodynamic studies, which quantify drug efficacy indicators such as cough inhibition and inflammation inhibition according to the level of levodropropizine plasma exposure, the relationship between the level of plasma exposure to levodropropizine and its therapeutic benefit may become clearer.

Additionally, the significant positive correlation between the V/F of levodropropizine and BSA found in this study implied that the level of plasma exposure to levodropropizine would be reduced (with high distribution to peripheral tissues) in individuals with relatively large BSA, even if dosage regimen to levodropropizine was the same. Although this study was based on pharmacokinetic results obtained in a healthy adult population, it was interesting to explore significant correlations between eosinophil levels, BSA, and pharmacokinetic parameters. This is because the healthy adult population has physiochemical parameter values that fall within a relatively narrow range of normal. Therefore, the significant correlations between eosinophil levels, BSA, and pharmacokinetic parameters explored in this study suggest they may possibly be expanded to interpret a wide range of clinical data including patient populations and/or various age groups in the future. For example, based on the search for potential candidate covariates explored in the results of this study, if levodropropizine is administered to patients with respiratory diseases with increased eosinophil levels, it may have advantages in terms of pharmacokinetics (with high plasma exposure levels) and produce a relatively excellent therapeutic effect. In addition, it may be implied that the dosage of levodropropizine needs to be relatively increased in obese subjects with high BSA to observe the same therapeutic effect as in subjects with low BSA. These are expanded interpretations of what covariates could mean in terms of pharmacokinetics-related therapeutics based on correlation trends between the candidate covariates explored in this study and levodropropizine pharmacokinetics. In addition, it is suggested that in future studies related to levodropropizine, it will be necessary to conduct dosage settings and drug efficacy studies that take into account the covariate correlations obtained in this study.

As previously mentioned, levodropropizine is contraindicated in individuals with severe liver dysfunction. Additionally, careful consideration is required when administering levodropropizine to individuals with impaired renal function. This clearly indirectly hints at the possibility that liver and/or renal functions may affect the pharmacokinetics of levodropropizine, resulting in changes in efficacy and toxicity. However, in this study, the final search for indices related to liver function and renal function as candidate covariates in relation to the interpretation of the pharmacokinetic diversity of levodropropizine failed. This may be related to the fact that this clinical study was conducted on a group of healthy adults, and most of the liver and renal function-related indicators in all subjects were within normal ranges. As a result, when interpreted based on the results of this study, it was implied that when levodropropizine is administered orally to a healthy adult population, the effects of liver and/or renal function on pharmacokinetic variability between individuals can almost be ignored. In other words, consideration of liver and renal function factors will not be very important in the clinical application of levodropropizine to the healthy adult population. The failure to explore candidate covariates of liver function indices in explaining inter-individual variation in levodropropizine pharmacokinetics does not contradict the interpretation of the association of levodropropizine’s metabolic effects with eosinophil levels explored in this study. The temporary decrease in the hepatic metabolism of levodropropizine due to increased eosinophil levels does not directly indicate liver damage; rather, it may be more appropriate to interpret it as a temporary inhibition of hepatic metabolic enzyme functions related to inflammatory factors associated with increased eosinophil levels. As a result, the factors targeted as liver function indicators in this study mainly represent the degree of liver damage, and eosinophil levels were interpreted to be a more sensitive indicator as a suitable factor to be associated with temporary decreases in the metabolism of levodropropizine.

Although the results of the target liver function panel presented in this study (Figure 2) were single-exposure results and were conducted in healthy adults, this was the first confirmed report with specific experimental values for toxicity and in vivo safety. The lack of notable changes in target liver function panel values (with no other significant side effects) before and after the levodropropizine clinical trial period suggested sufficient safety of levodropropizine, at least in healthy populations with normal liver function. In the future, in vivo safety evaluation will need to be further confirmed by conducting clinical trials of levodropropizine based on multiple exposures and/or in various groups (such as infants, the elderly, and patients with liver disease).

## 4. Methods

### 4.1. Study Workflow

This study was conducted in four main steps and began with a bioequivalence study following a single oral dose of levodropropizine. In the bioequivalence test, only the in vivo results for the reference formulation, not the test formulation, were used. This is because the main purpose of this study was not to determine bioequivalence between the test and reference formulations, but to confirm the pharmacokinetic characteristics and inter-individual variability of levodropropizine, which is already on the market. As a first step, levodropropizine concentrations (according to time after administration) in participants’ plasma samples and physiochemical parameters (derived from blood analysis) were determined using ultra-performance liquid chromatography–mass spectrometry (UPLC-MS/MS) and serological analysis instruments (operated by reflectance spectrophotometry), respectively. In the second step, parameter calculation was performed through non-compartmental analysis using the pharmacokinetic results based on levodropropizine plasma concentrations determined in the first step. Additionally, the calculated pharmacokinetic parameter values were used to search for potential valid covariates through correlation analysis with physiochemical parameter values for each individual. As a third step, compartmental pharmacokinetic model screening of levodropropizine was performed. This was to confirm the appropriate basic structure for future expansion of advanced levodropropizine population pharmacometrics modeling and to check whether the candidate covariates explored in this study could be equally applied to a structured model. As a fourth step, analyses of toxicity and clinical safety aspects of levodropropizine were performed. This was to check for any toxicity of levodropropizine that may occur during clinical application.

### 4.2. Bioequivalence Test

A bioequivalence study on levodropropizine was conducted on 40 healthy Korean men. This clinical trial was conducted as open-label, randomized, fasting (more than 10 h before administration), and single oral dose. Only the reference formulation (Dropizin Tab., Kolon Pharmaceutical, Gwacheon-si, Republic of Korea) data from a two-by-two crossover study with a 7-day washout period were used for pharmacokinetic analysis (as a retrospective approach).

#### 4.2.1. Subjects

All participants had eligible diagnostic test results at screening for the final decision to participate in the clinical trial. That is, the age at the time of screening was 19 years or older, BMI (as an obesity index) was within the range of 18–30 kg/m^2^, and the results of diagnostic tests (blood and urine tests) and electrocardiogram were within normal values. And in the drug history test, no concomitant use (within 30 days before test date) of drugs that could affect this clinical trial (such as inducing and/or inhibiting metabolic enzymes) was confirmed. All participants had no clinically significant congenital or chronic diseases and no history of gastrointestinal resection that could affect drug absorption. It was also confirmed that the participants had not consumed excessive alcohol within 1 month prior to the test date and had no history of hypersensitivity to levodropropizine and/or phenylpiperazines (as the core structure of levodropropizine). All participants provided written informed consent to conduct bioequivalence and pharmacokinetic studies. The safety of participants was monitored by recording adverse events that occurred during the study, checking vital signs, and diagnostic tests during and after the study. This clinical study was conducted in accordance with the revised Declaration of Helsinki for Biomedical Research Involving Human Subjects and Good Clinical Practice. The entire process from the approval (Trial No. 100744; officially approved by the Ministry of Food and Drug Safety [MFDS; Cheongju-si, Republic of Korea] on 1 November 2022) to the execution of this clinical trial was thoroughly monitored by the MFDS.

#### 4.2.2. Sampling

For each subject, pre-dose (0 h) and post-dose 0.08 (5 min), 0.17 (10 min), 0.33 (20 min), 0.50 (30 min), 0.75 (45 min), 1, 1.5, 2, 3, 4, 6, 8, and 12 h (14 times in total) blood draws were performed by placing a heparin (150 unit/mL)-locked angio-catheter (JELCO 22G, Smiths Medical, Minneapolis, MN, USA) in a vein in the back of the subject’s arm or hand beginning at approximately 7:00 a.m. on the day of the study and collecting approximately 6 mL of pre-dose blood (as a blank sample). At the time of blood collection, approximately 1 mL of blood was preemptively discarded to remove the saline remaining in the collection set, and then approximately 6 mL of blood was collected into a vacutainer (Becton Dickinson, Franklin Lakes, NJ, USA) containing sodium heparin, and then injected with heparinized saline to prevent clotting of the blood remaining in the catheter. The collected blood was centrifuged at 3000× *g* for 10 min with setting to 4 °C and approximately 1 mL of plasma was taken and transferred to Eppendorf tubes (Hamburg, Germany) and stored in a deep-freezer set at approximately −80 °C until sample quantitative analysis.

#### 4.2.3. Determination of Clinical Physiochemical Parameters

Analysis of physiochemical parameters was performed to identify effective factors that can explain the inter-individual pharmacokinetic variation of levodropropizine and to confirm toxicity and safety that may occur during the test. BSA was calculated using the Mosteller equation [25] and BMI was calculated using the Kaup index [26]. Hematological analysis was performed using a dry automated analyzer, microsides VITROS (Ortho Clinical Diagnostics, Raritan, NJ, USA). eGFR and CrCL were calculated using Modification of Diet in Renal Disease (MDRD) [27] and Cockcroft–Gault equations [28], respectively.

### 4.3. Determination of Plasma Levodropropizine Concentrations

Quantification of the concentration of levodropropizine in plasma samples was performed using UPLC-MS/MS optimized with reference to the previous analytical method report [14] and was fully verified according to Food and Drug Administration guidance on bioanalytical method validation [29]. The analysis systems consisted of a Xevo^TM^ TQ-XS triple quadrupole mass spectrometer (Waters Corp., Milford, MA, USA), and the stationary phase was a C_18_ column (2.0 × 100 mm, 3 μm; Unison UK-C18, Imtakt Corp., Portland, OR, USA). The mobile phases selected were 5 mM aqueous ammonium format and acetonitrile, and the composition ratio was 70:30 (*v*/*v*). The internal standard method was applied to the quantification process of levodropropizine, and levodropropizine-d_8_ (as a structural analogue of levodropropizine) was used as the internal standard. The calibration curve in the plasma matrix of levodropropizine was obtained from the concentration range of 5 to 1000 ng/mL. The selected parent ions in mass spectrometry of levodropropizine and levodropropizine-d_8_ were 237.10 and 245.20 (*m*/*z*), respectively, and the targeted daughter ions were the same at 119.90 (*m*/*z*). Quantification of analytes was performed via positive electrospray using multiple reaction monitoring mode. The retention times of levodropropizine and levodropropizine-d_8_ in the column were similar, approximately 1.30 min. Method validations were performed for selectivity, sensitivity, accuracy, precision, carryover, stability, and matrix effect.

### 4.4. Safety Test

The drug safety information for levodropropizine [30] includes precautions for use in individuals with hepatic impairment. In other words, the administration of levodropropizine is prohibited for people with severe hepatic impairment. This indirectly suggested changes in pharmacokinetics related to drug metabolism of levodropropizine and/or the possibility of hepatotoxicity. Therefore, as part of drug safety screening and evaluation (according to levodropropizine exposure), liver function tests were performed, and AST, ALT, and γ-GTP were set as target monitoring factors. AST and ALT are transamination enzymes present in hepatocytes, and γ-GTP is an enzyme involved in liver detoxification, and both were indicators of liver damage [31]. Safety screening and assessment were performed at the last point of sampling, 12 h after drug administration.

### 4.5. Pharmacokinetic Analysis

The basic pharmacokinetic parameters of levodropropizine were analyzed by non-compartmental analysis using Phoenix WinNonlin software (version 8.3; Certara Inc., Princeton, NJ, USA). The AUC_0-inf_ was calculated as the sum of AUC_0-t_ and C_last_/k, where C_last_ was the last measurable concentration, t was the time of C_last_, and k was the elimination rate constant for the terminal phase. AUC_0-t_ was calculated using a linear up–log down method (generally apply linear and log trapezoidal rules before and after T_max_, respectively) from 0 to t h after oral administration. This is because, as a result of screening the basic model structure in this study, two compartments (with first-order absorption) were suitable, so it would be better to apply the log trapezoidal rule to calculate the area under the profile curve in the distribution and dissipation phase after T_max_. The MRT was estimated as the ratio of area under the first-order moment curve (AUMC) and AUC_0-inf_, where AUMC was calculated as the area of the graph under the product of time and concentration over time. T_1/2_ was calculated as 0.693/k and V/F was calculated as dose/k × AUC_0-inf_. The CL/F was calculated by dividing the dose of levodropropizine (60 mg) by AUC_0-inf_, where F is the bioavailability of oral administration. From the plasma levodropropizine concentration–time curve for each individual, C_max_ and T_max_ were determined. All pharmacokinetic parameter values were estimated as mean ± standard deviation.

### 4.6. Exploring the Basic Structure of the Model

In this study, a screening of the basic structure of a suitable pharmacokinetic model of levodropropizine was additionally performed. This process involved the naïve pooled method. Several criteria were used to guide model selection. The AIC was used to assess the balance between model accuracy and complexity, with lower AIC values suggesting more favorable models. The SC was also applied, similar to AIC, to evaluate model fit and complexity while imposing a more stringent penalty on the number of model parameters, thus favoring simpler models. The WSSR was calculated to quantify the model’s error, defined as the sum of squared deviations between predicted and observed values, where lower values indicate a better model fit. The model’s numerical stability was gauged using the condition number, with higher values suggesting increased sensitivity of parameter estimates to errors. Lastly, the run value, reflecting the number of instances a data point falls above or below the model’s predictions [32], was considered to evaluate model consistency and random variability within the data. Generally, a higher run value is indicative of a model that accurately fits the data. And from a qualitative perspective, the model’s suitability was reconfirmed through the model fitting results of the observations.

### 4.7. Approach to Covariate Exploration

Correlation analysis between pharmacokinetic parameter values calculated through non-compartmental analysis and physiological and biochemical parameters was performed using Seaborn library in Python (version 3.12.1). Seaborn was a visualization library for plotting statistical graphics in Python. The main tools used in this process were heatmap analysis and significant linear regression analysis between two independent variables. In the correlation analysis, physiological and biochemical parameters as well as pharmacokinetic parameters were all treated as continuous data.

First, significant correlation factors were derived through heatmap analysis reflecting both pharmacokinetic parameters and physiological and biochemical parameters. In the heatmap analysis results, correlation coefficients ranged from −1 to +1, with higher positive correlations visualized in red and higher negative correlations visualized in blue. As a result of the heatmap analysis, it was determined that if the correlation coefficients were greater than +0.3 or less than −0.3, there would be a valid positive and negative correlation, respectively [16]. The correlation factors derived through heatmap analysis were re-judged for effectiveness between pharmacokinetic and physiochemical parameters using *R*-squared, *F*-statistic, and *p*-value based on linear regression. *R*-squared, a measure between 0 and 1, indicates how well the regression model explains variability in the data, with higher values indicating better model fit. *F*-statistics are used to test the overall fit of the model by evaluating the collective contribution of the variables to the explanatory power of the model. *p*-value is used in statistical hypothesis testing to determine whether the effects of correlation factors are due to chance or are statistically significant. Typically, a *p*-value of less than 0.05 was considered indicative of significant effects. As a result, pharmacokinetic parameters and physiochemical factors with an *R*-squared value greater than 0.09 [16] and a *p*-value less than 0.05 were selected as effective candidates for explaining the inter-individual pharmacokinetic variations of levodropropizine. And among the selected valid candidates, those whose correlational tendencies could be theoretically explained were interpreted as the final core covariates.

## 5. Conclusions

In this study, BSA and eosinophil levels were explored as effective candidate covariates that could explain the inter-individual pharmacokinetic variability of levodropropizine. An interesting point among the covariates explored was the significant positive correlation between eosinophil levels and plasma exposure to levodropropizine. Increased plasma exposure to levodropropizine in individuals with increased eosinophil levels was interpreted as a factor that may improve the efficacy of levodropropizine based on pharmacokinetic benefits in the treatment of indications such as eosinophilic asthma. This study presents a useful covariate perspective in interpreting the previously limited pharmacokinetic diversity of levodropropizine and suggests new causal advantages of selecting levodropropizine in clinical-indication-based pharmacotherapy.

## Figures and Tables

**Figure 1 pharmaceuticals-17-00234-f001:**
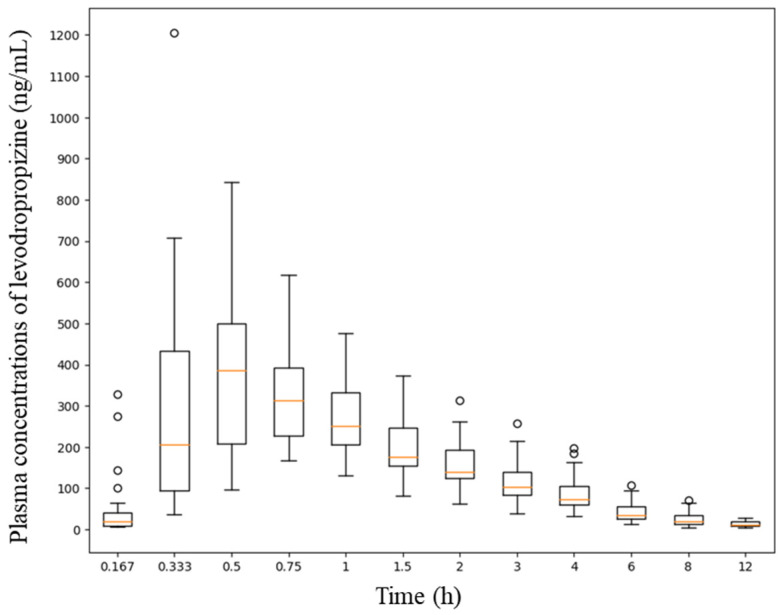
Plasma concentration profiles (depicted as a boxplot) following single oral exposure to levodropropizine (60 mg tablet) in 40 healthy Korean men. The X- and Y-axes in the graph represent time and plasma concentration values according to levodropropizine exposure, respectively.

**Figure 2 pharmaceuticals-17-00234-f002:**
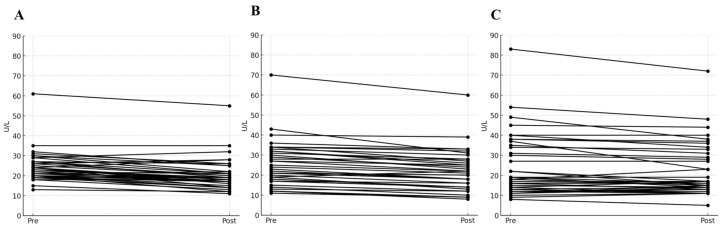
Spaghetti plot of changes in aspartate transaminase (**A**), alanine transaminase (**B**), and gamma-glutamyl transpeptidase (**C**) levels before (0 h) and after (12 h after dose) single oral exposure to levodropropizine. Pre and Post on the *X*-axis in the figure refer to 0 and 12 h after oral exposure to levodropropizine, respectively.

**Figure 3 pharmaceuticals-17-00234-f003:**
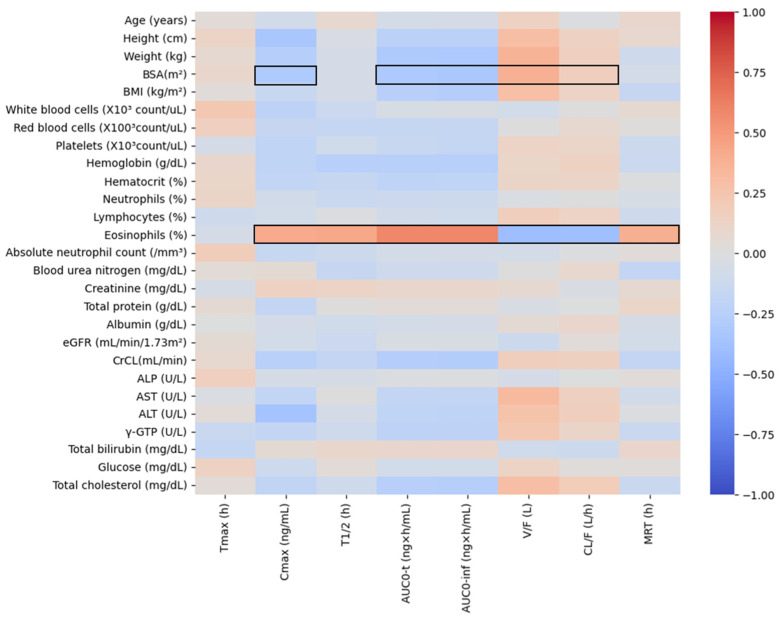
Heatmap results for correlation analysis between pharmacokinetic parameter values according to oral administration of levodropropizine (60 mg tablet) and physiochemical parameters of each individual. Positive correlations are colored red and negative correlations are colored blue. Black line boxes displayed in the heatmap indicate the detection of factors that can reasonably explain the correlation between physiochemical and pharmacokinetic parameters with an absolute correlation coefficient of 0.3 or higher. BSA, body surface area; BMI, body mass index; eGFR, estimated glomerular filtration rate; CrCL, creatinine clearance; ALP, alkaline phosphatase; AST, aspartate transaminase; ALT, alanine transaminase; γ-GTP, gamma-glutamyl transpeptidase; C_max_, maximum plasma concentration; T_max_, time to reach C_max_; T_1/2_, elimination half-life; AUC_0-t_, area under the curve from 0 to observed time after administration; AUC_0-inf_, area under the curve from 0 to infinity time after administration; V/F, volume of distribution; CL/F, clearance; MRT, mean residence time.

**Figure 4 pharmaceuticals-17-00234-f004:**
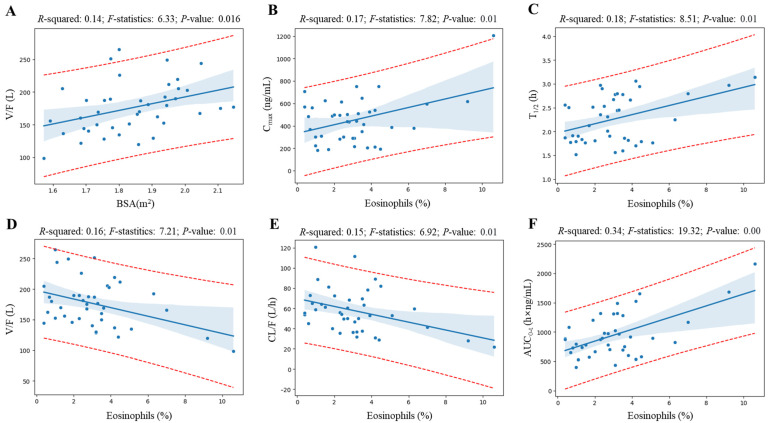
Linear regression analysis results for elements whose coefficient values are at the top of the valid correlation derived from the heatmap screening results. Blue dots, blue thick solid lines, and blue shading in the graph represent observations, the average line by the regression model, and the 95% confidence interval, respectively. The red dotted lines in the graph represent the 95% prediction interval. Correlations are shown for BSA–volume of distribution (V/F) (**A**), eosinophil level–maximum plasma concentration (C_max_) (**B**), eosinophil level–elimination half-life (T_1/2_) (**C**), eosinophil level–V/F (**D**), eosinophil level–clearance (CL/F) (**E**), and eosinophil level–area under the curve from 0 to observed time after administration (AUC_0-t_) (**F**), respectively.

**Figure 5 pharmaceuticals-17-00234-f005:**
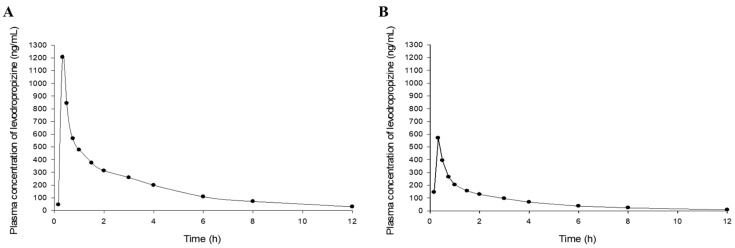
Graphic comparison of levodropropizine pharmacokinetic profiles (following single oral exposure to levodropropizine 60 mg tablet) between subjects with the highest (10.6%, (**A**)) and lowest (0.4%, (**B**)) eosinophil levels in this clinical study’s population.

**Figure 6 pharmaceuticals-17-00234-f006:**
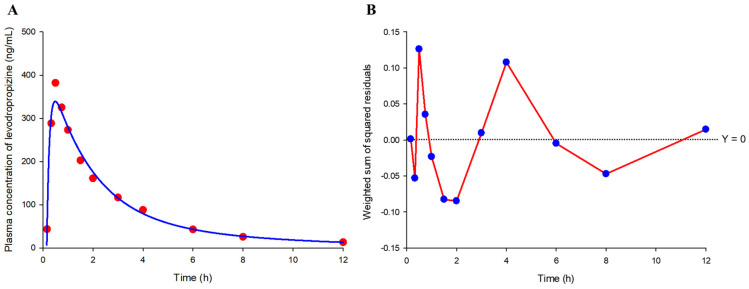
Model fitting results for observed mean plasma concentration values (**A**) and weighted sum of squared residual (WSSR) distribution patterns (**B**). The *X*-axis of the graph represents time after single exposure to levodropropizine 60 mg tablet. Red dots and solid blue line in the model fitting graph (**A**) represent observations and model predictions, respectively. The blue dots and solid red lines in the WSSR distribution pattern graph (**B**) represent WSSR values at each time point and the run results (based on 0) according to the connection of all points, respectively.

**Figure 7 pharmaceuticals-17-00234-f007:**
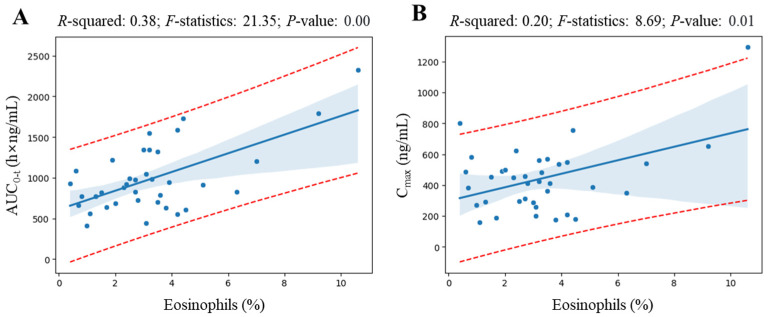
Linear regression analysis results between eosinophil levels and pharmacokinetic parameters (area under the curve from 0 to observed time after administration (AUC_0-t_) (**A**) and maximum plasma concentration (C_max_) (**B**)) derived from the basic structure of the explored compartment model (two-compartment model with a lag time). Blue dots, thick solid blue lines, and blue shading in the graph represent observations, the average line obtained in the regression model, and the 95% confidence interval, respectively. The red dotted lines in the graph represent the 95% prediction interval.

## Data Availability

All data and related materials are accessible in this manuscript.

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
