# Peer review of "Pharmacokinetic Analysis of Levodropropizine and Its Potential Therapeutic Advantages Considering Eosinophil Levels and Clinical Indications"

_pharmaceuticals, 2024, doi:10.3390/ph17020234_

Round 1

Reviewer 1 Report

Comments and Suggestions for Authors

The authors investigated effective covariates that explain the inter-individual pharmacokinetic variability of levodropropizine and revealed BSA and eosinophil levels in healthy population. The work is interesting but there are some critical issues in this work.

1.       Introduction is too long. The authors should describe it briefly.

2.       L69-71: The authors should describe dosages in in vivo study in detail.

3.       Figure 4: Although p values was less than 0.05, observations more than half were out of rage of 95% CI. It is not good model. The authors should reanalyze it or describe the reasons why most points is out-of-range.

4.       Figure 5: Two cases with the highest and lowest eosinophil levels was compared but there was no information such as dosage and organ functions in the cases. The authors should describe compare data on patient’s background between the cases.

5.       L254: Figure on the weighted sum of squared residuals (WSSR) need to be shown.

6.       L256: Results on the runs test need to be described. Ex, How many successes? etc.

7.       Discussion: The authors mentioned association between eosinophil and hepatic functions. Why did not hepatic functions include in the model? It has antinomy.

8.       4.2.3.: Physiochemical parameters need to be described in the section of Results. Moreover, the data had better be shown as a table as well.

Author Response

We have attached a point-by-point response to reviewer comments as a word file.

Please check the attached file (file name: Point-by-point response).

We used red font (with underline) instead of black to indicate the revised or added portions in our manuscript (at ‘Manuscript (R1)’ file).

The answers to reviewer comments were described in red font under each question (at ‘Point-by-point response’ file).

Reviewer 2 Report

Comments and Suggestions for Authors

The paper addresses the pharmacokinetics of levodropropizine in healthy subjects with the aim of conducting a bioequivalence test. However, the presentation of the results, particularly those from the bioequivalence test, lacks clarity. The overall organization of the results and the subsequent discussion also need improvement. Despite containing valuable new data, the manuscript requires significant revision before publication. A suggestion is to concentrate on refining the presentation of pharmacokinetic results and exploring correlations with Body Surface Area (BSA) and eosinophil counts for a more focused and coherent manuscript.

The title can be revised from “Presenting the potential therapeutic advantages of levodropropizine considering eosinophil levels and clinical indications: A pharmacokinetic analysis-based approach” to “Pharmacokinetic analysis of levodropropizine and its potential therapeutic advantages considering eosinophil levels and clinical indications”

Abstract

The abstract can be modified to emphasize the impact of tested co-variates on the pharmacokinetics of levodropropizine.

Introduction

The introduction should be condensed to two-thirds of its current length.

Lines 72-77: Combine the information into a single sentence, for example: "Optimizing cough treatment, particularly with levodropropizine, involves applying personalized medicine principles."

Lines 77-85: These lines can be omitted.

To enhance the introduction, consider removing overly general statements or information that explains basic pharmacokinetic principles.

Results

Line 147: “Tmax was very fast” is better to be replaced by “Tmax was very short”

Section 2.2. Safety screening has to be placed at the beginning of the results. This approach can ensure that readers are aware of any safety concerns before dealing with the pharmacokinetic details. If the authors think that it could be more appropriate, they can place this information at the end, after the presentation of the results about pharmacokinetics.

Is it essential to compare non-compartmental analysis with the naive pooled method in the following sentence: “All correlations were statistically significant (P-value lower than 0.05) and overall were higher than the correlations (higher R-squared) between pharmacokinetic parameters (Cmax and AUC0-t) calculated by non-compartmental analysis and eosinophil levels”? What is the additive value?

Discussion

In the second paragraph of the discussion, the role of eosinophils was elucidated; however, the direct impact on the pharmacokinetics and pharmacodynamics of the antitussive drug remains unclear.

What is more relevant, to have higher concentrations of levodropropizine in plasma or at the site of action? It has to be stated that higher count of eosinophils is a predisposition for slower elimination due to inhibited metabolism.

Lines 334-340: The conclusions are very strong and this study does not allow to suggest the changes in the doses.

Materials and methods

Lines 367-371: Why a question about bioequivalence has been added? It is not essential. The results from the bioequivalence study were not well presented.

Line 489: AUC0-t has to be calculated according to linear up-log down method

Author Response

(The authors gave the same response as above.)

Round 2

Reviewer 1 Report

Comments and Suggestions for Authors

The authors revised appropriately. No further correction is necessary.

Reviewer 2 Report

Comments and Suggestions for Authors

The authors revised the manuscript according to the reviewers' remarks. Additional explanations were added to the Materials and Methods section so that readers can completely understand the procedures. Satisfactory reasons were provided for instances where certain remarks were not addressed.  The manuscript can be accepted for publication.